# The use of social media as a two-way mirror for narcissistic adolescents from Austria, Belgium, South-Korea, and Spain

**Lluís Mas Manchón** [1]* , **David Badajoz Dávila** [2]

**1** Department of Communication, Universitat Pompeu Fabra, Barcelona, Spain, **2** Department of Advertising, PR, and Audiovisual Communication, Universitat Autònoma de Barcelona, Barcelona, Spain

☯ These authors contributed equally to this work.
* Lluis.mas@upf.edu

## Abstract

The use of social networking sites (SNS or social media) often comes with strong self-centered behaviors to promote self-appearance. The relationship between narcissism and social media use has intensively occupied scholars in the last decade, yet not much research has focused on, first, how the intensity of social media use (SNS use) is associated with narcissism through a self-centered appearance focused use of these SNS; and second, whether these associations are moderated or not by cultural differences of the country of origin in such a critical age of personality formation and (global) culturalization as the transition from pre-adolescence to adolescence. We performed a correlation and mediation analysis on a cross-sectional survey among Austrian, Belgian, Spanish, and South Korean adolescents ($n = 1,983$; $M_{age}$ 14.41, 50.3% boys) examining the adolescents' daily usage of social media, their self-centered appearance focused behavior, and the reported narcissism. Findings show that a self-centered appearance focused use of SNS (SCA) moderates the association between SNS use and narcissism, especially for males from the three European countries. We have also particularly found that the years of use, number of friends and time spent in FB are associated with narcissism. Since SCA is defined in the study as narcissistic behavior in SNS, we argue that social media are part of the socialization process as both reinforcers and catalyzers of narcissism.

## Introduction

The use of social networking sites (SNS) by children and adolescents has increased steadily in the last decade. A study in 2016 by BBC found that more than 75% of children in the UK between 10 and 12 had social media accounts [1]. More recently, the Pew Research Center reported in 2018 that 95% of teens in the US (from 13 to 17) had access to smartphones, 72% were using Instagram, 69% Snapchat, 51% Facebook, and 45% admitted being online almost constantly [2]. Although there is a great renewal and variability in the specific platforms used by each generation -in 2021, TikTok or Twitch are the fastest growing platforms amongst adolescents, most platforms are based on taking and sharing images and videos with a broad range

**Data Availability Statement:** All relevant data are within the paper and its Supporting information files.

**Funding:** The author(s) received no specific funding for this work.

**Competing interests:** The authors have declared that no competing interests exist.

of formats, technical manipulations and uses. In general, these platforms permit great visibility, social feedback, close control over the image, and easy accessibility by the users [3, 4].

The use of SNS often comes with strong body surveillance and body exhibition practices, which can be associated with several body image and personality risks [5]. Narcissism is a prevalent personality disorder amongst youngsters associated with the own's body promotion and attention-seeking [6]. The relationship between narcissism and the use of social media has intensively occupied scholars in the field during the last decade. Two meta-analytic reviews have stablished a strong and consistent association between narcissism and social media use, particularly self-centered behavior such as posting selfies or status updates, but also quantitative factors such as time spent on social media or number of friends [7, 8].

However, some works cast doubt on the direct relationship between social media use (SNS use) and narcissism [9]. Narcissism is conceptualized as a multifactorial personality trait that develops along life due to psychosocial and cultural factors [10, 11]. Yet not much research has focused on how factors such as the type of social media use, culture or gender can moderate narcissism. Since personality is molded by an interplay of crucial cultural and social factors such as education, the nation, or the media [12], this research tries to shed light on how social media behavior is associated with narcissism, and to what extent national culture and gender moderate theses associations during adolescence as a crucial period of personality formation [13, 14]. Importantly, narcissism's subcomponents exhibitionism, exploitativeness and entitlement are particularly prevalent in early ages [15]. While the western-oriented pervasiveness of SNS add to the traditional globalization dynamics as homogenizing factors towards global culture [16] and the way citizens use SNS, local and national cultures can explain important differences of social behavior and personality such as is the case of the relationship between individualist versus collectivist cultures and high versus low narcissism [17–19].

Therefore, the objective of this research is to explore how the intensity of SNS use is associated with narcissism through a self-centered appearance focused use of SNS; and second, whether these associations are moderated or not by cultural differences and gender during preadolescence and the early years of adolescence.

## The relationship between social media use and narcissism

### Intensity and self-appearance as narcissism indicators of social media use

Narcissism was conceptualized in 1988 as 'a grandiose sense of self-importance or uniqueness; a preoccupation with (. . .) power, beauty, or ideal love; exhibitionism; (. . .) interpersonal exploitativeness, relationships that alternate between extremes of overidealization and devaluation; and a lack of empathy' [20]. Narcissism has usually been defined in three dimensions (leadership/authority; entitlement/exploitativeness and grandiose exhibitionism [21, 22]. Vazire [23] posed that narcissists are more likely to behave in an exhibitionistic way, seek attention from others and focus on physical appearance. More specifically, narcissism personality trait is associated with showing the body socially, control and constantly change appearance and use strategies to promote oneself and attract other's views, approval, or appreciation [3, 24–26]. Those that to some extent behave with great exhibitionism cannot stand being ignored and need to promote themselves and gain interest and attention from the others [24, 27].

Narcissism is a personality trait that develops in early childhood and adolescence, that is when cultural impacts are more powerful. Adolescents are defined as children and young adults of ages between 10 and 19 who are still developing their identity, and therefore, are exposed to cultural changes [28]. For adolescents there is a special "risk of adolescent cultural identity confusion" as well as "internalizing and externalizing pathological behaviors" [16].

Social networking sites (SNS) are online platforms in which users continuously generate and share images and text with known and unknown peers. Facebook (FB henceforth) is one of the first and is certainly the most popular SNS worldwide with 1.8 billion daily users and 2.8 billion monthly users [29], but there are many others attracting the interest of youngsters around the world such as Instagram, Tik Tok, Pinterest, Periscope, or Twitch, to name a few. All of those are image-based tools to share content with friends and contacts. Some are even used as personal photo albums accessible to a great number of friends and contacts. Thus, social relationships are partially, and in some cases exclusively, mediated by the images shown of ourselves. As observed at the dawn of FB, this tool was being used by adolescents to build a hedonistic image based mostly on a visual identity of their 'hoped-for possible' [30]. Seminal work by Buffardi and Campbell [31] showed SNS as a potentially suitable arena for narcissistic self-regulation as these tools allow a close self-control of the information provided, especially images, and a large network of superficial relationships. Walters and Horton [32] conducted a study on the direction of the effects of FB use over personality, narcissism and, specifically, grandiose exhibitionism. In sum, both the nature of SNS and the mechanisms to use SNS align well with narcissism behavior, thus using social media and narcissism feed into each other.

Yet, the relationship between social media use (SNS use) and narcissism remains unclear [33]. To guide this discussion, scholars have focused on the type of social media use such as an agentic versus a communal use of social media [5, 32, 34, 35]. An agentic use of SNS is based on the adolescents' concern on the importance of attractiveness in SNS; and manifests in behaviors such as constantly taking and publishing selfies, updating status, changing profile picture, using techniques to look better in pictures, removing a tag from a picture in which one considers not to be attractive enough, comparing systematically pictures with others, or follow-up one's pictures comments [36]. A communal use of SNS would focus on actions towards others such as liking others' posts, private messaging, birthday wishes or reading friends' posts and making comments [35]. However, research on how the intensity of social media use relate to narcissism is far from conclusive:

H1: A higher social networks use will be related positively with narcissism.

Following the rationale of this hypothesis, the explanatory mechanism for social media to be inextricably connected with narcissism is the social media determinism towards a self-centered image-based use, which aligns well with the reinforcing effects spirals model [37]. In other words, the more anyone uses social media in general, the more likely his/her behavior will be agentic and focused on the image, thus an increasing use of social media inevitably leads to an increasing self-centered appearance focused use:

H2: A higher social media use will be related with higher self-centered appearance focused use of social media.

Further, we draw on the well-reported connection between self-centered appearance focused social media use and narcissism [38]. Two dimensions of a self-focused type of SNS use may interact with narcissism. First, high levels of narcissism are related with the tendency to apply attractiveness and appearance as the main criteria to post pictures [20, 23, 26, 31, 39–43]. Particularly, studies have found a correlation between narcissism (self-sufficiency, vanity, leadership, admiration demand or grandiose exhibitionism) and the frequency of posting selfies based on attractiveness [40, 44–46] or valuing pictures for their physical attractiveness [24, 26]. Siibak [47] found that the profile picture is used by youngsters between 11 and 18 to construct their ideal attractive self or the ought-self, and that girls tended to prioritize their aesthetic, emotional and self-reflecting dimensions in these images.

Second, narcissists are constantly using SNS to compare this physical appearance with their peers. Narcissists apply self-regulatory strategies to affirm an unrealistic positive self-concept [48] and have no interest in strong interpersonal relationships [5]. Toma and Hancock [49] found that self-affirmation is the most important outcome of using FB. And Seidman's study on undergraduates [50] found out that self-presentation behaviors in FB (self-promotion and the tendency to show ideals in particular) were predicted by low conscientiousness and high neuroticism. Carpenter [24] and Mehdizadeh [26] found that grandiose exhibitionism predicted self-promoting behaviors such as accepting strangers as friends and constantly updating the status, profile picture, and pictures in general. And Ong et al. [5] found that narcissism is a predictor of self-generated content in 11–16-year-old adolescents.

Thus, we posit that these two dimensions of a self-centered appearance focused use of SNS–how attractive is physical appearance and self-regulatory mechanisms based on a systematic comparison with peers- are both indicators and reinforcers of a narcissistic personality:

H3: Self-centered appearance focused use of social media correlates positively with narcissism.

## The relevance of country and gender on a narcissistic use of social media

As a personality trait, narcissism stems from culture [10]. Culture is conceived as a sum of environmental factors such as language, tradition, values, artifacts, or media messages [12]. Many of these factors overlap across cultures, which is why countries cannot be fully equated with cultures. 'Globalization' is commonly defined as the worldwide spread of people, goods and ideas across borders and countries [51–53]. Globalization exposes humans to multiple new cultures and the outcomes can be perceived as positive (increased creativity, less prejudice) or as negative (culture loss, ethnic bias, or identity crisis) [54]. Globalization reached a new dimension with the outburst of technological networks and social media because a great portion of humanity can individually and virtually communicate freely and in real time with any one from anywhere.

Hermans and Dimaggio [55] suggested a connection between globalization and the "identity disturbances" observed since the 1980s. According to Albrow [56] globalization may even alter the "place in social structure and culture" of different age groups in the same culture, particularly young segments of society [16, 55]. Thus, age has traditionally been conceived as a homogenizing factor across cultures. McCrae et al. [57] found consistent personality similarities between people of the same age span from Germany, Italy, Portugal, Croatia, and South Korea, and consistent differences between youngsters and middle adulthood participants from these countries–the former being more open and extraverts and the latter being more agreeable and conscious.

Yet, countries or group of countries have been taken and still are taken as solid cultural divisions [18, 58]. Culture studies have a long tradition in classifying countries as more individualistic or more collectivistic [19] among other dimensions [58]. Individualists place greater interest in oneself and their independence, whereas collectivists are concerned about the interests of the group (family, organization, society. . .). As Grijalva and Newman's study [17] suggest, collectivist cultures weaken behaviors against the group, and McCrae and Terracciano [18] found that individualistic cultures (namely American and Europeans) scored higher in personality trait extraversion than collectivist cultures (Africans and Asians). Coherently, high individualist nations score higher in the enhancement of individual traits whereas low individualist nations score higher in enhancement of communal traits [59].

A consistent thread of studies has reported the existence of both a cross-cultural narcissism as well as some country-based narcissism traits, lately giving more importance to individual

narcissism [60], but there are very few accounts of cross-cultural similarities or differences in narcissism as related with the social media activity and usage. In general, studies have found no such significant and theoretically relevant differences in multiple comparisons of western and eastern countries [45]. In addition, the individualist-collectivist division does not successfully explain some minor cross-cultural differences in the social media usage. For instance, the study by Errasti et al. [61] showed that Thai adolescents scored higher in emotional disclosure and expression in social media than Spanish adolescents. This finding would oppose to the expected higher online emotional expression of participants from the country with higher individualism -Spain- to fill the comparatively lower offline social and emotional bounds. Besides, in this study, Spanish adolescents scored higher in leadership than Thai counterparts, but no clear differential Facebook usage was found.

Further, the intersection between gender studies and culture studies has usually evidenced overlapping or contradictory stereotypes [62]. Gender differences occur across countries despite gender egalitarian policies [62]. For instance, in Costa et al.'s study [63] women scored higher in "Neuroticism, Agreeableness, Warmth, and Openness to Feelings, whereas men were higher in Assertiveness and Openness to Ideas" (p. 322), especially in western (individualistic) cultures such as the US and Europe. Further, studies have reported that men score higher in narcissism (especially Exploitative/Entitlement and Leadership/Authority, and to a lesser extent GE) than women [15, 64].

Again, not much research has found consistent gender differences between social media usage and narcissism. Arpaci et al. [12] found that the correlation between narcissism and selfie posting behavior was significant for men only, while females were using social media more intensively. Errasti et al. [61] reported a higher FB use by females from Spain and Thailand, but no particular connection between this social media use and narcissism behavior. Their study did find that Spanish males score higher in social media behavior indicating exhibitionism than females, and the opposite in the case of the Thailand sample. In line with this, Kim and Jang [65] found a correlation between social media use frequency and narcissism for males but not for females; and, importantly, the motivation for self-presentation in SNS correlate with narcissism for both men and women.

Overall, there is no conclusive research analyzing country and gender differences in how social media use relates with narcissism, hence we pose the research question:

RQ: What are the country and gender variations in the relationship between social media use (SNS use) and narcissism (H1), SNS use and self-centered appearance focused use (SCA) of SNS (H2), and SCA and narcissism (H3)?

## Narcissism and the extended use of Facebook

Previous hypotheses are grounded on the reinforcing cumulative effects of social media use and narcissistic personality in different countries. If the two ends -SNS use and narcissism-feed into each other, then an increasing use of social media is related with being narcissistic (H1) and with a narcissistic use of social media (H2), which in turn relates with narcissism (H3) [66]. According to this spiral form, a moderate or decreasing use of social media would be associated with a moderate or decreasing narcissistic behavior. Simply put, social media are conceived as narcissistic tools, hence higher levels of activity in social media would to some extent be equated with higher levels of self-centered appearance-based behavior and narcissism, and the opposite for lower levels of activity.

Being the first global SNS, Facebook can be an insightful case of study for this model. First, most studies on social media include Facebook, hence most of the evidence provided applies

specifically to this platform. Second, although the use of FB by adolescents has decreased in some countries, it is still widely used in every country by all generations including adolescents. So, authors have found positive experimental correlations between the time spent in SNS and narcissism [7, 26, 35, 44]. Here we add the distinction between the reported active versus passive use of Facebook [43, 67] as connected with narcissism:

H4: An active (vs passive) Facebook use correlates positively with narcissism.

Further, interestingly, social media can substitute or even hinder social life. Authors have found a negative correlation between time spent in FB and the percentage of friends that users knew in person [35]. Indeed, there are cognitive constraints that limit the active relationships humans can hold -an average of 150, Dunbar's number, thus the number of friends in social media is expected to correlate negatively with the number of friends offline [68]. Having more friends in Facebook may be associated with more online activity and more time spent in Facebook, less social life or offline socialization, and perhaps more narcissism [35, 39]:

H5: The number of Facebook friends is positively related with narcissism.

A final hypothesis may show a preliminary connection between a long-term use of social media and narcissism. Being one of the oldest social media, Facebook is one of the few SNS that brings the opportunity to focus on the years of use as a variable:

H6: The number of years of Facebook use correlates with narcissism.

To add clarity to the theoretical stand of the article, the entire model of hypothesis is shown graphically in Fig 1.

## Methods

### Materials, participants, and procedure

A cross-sectional study was conducted between February and May 2017 in Austria, Belgium, Spain, and South Korea, called the Intercultural Study Project (ISP). The country to collect data is a variable to control country variations in the relationships posed in the hypothesis. The study

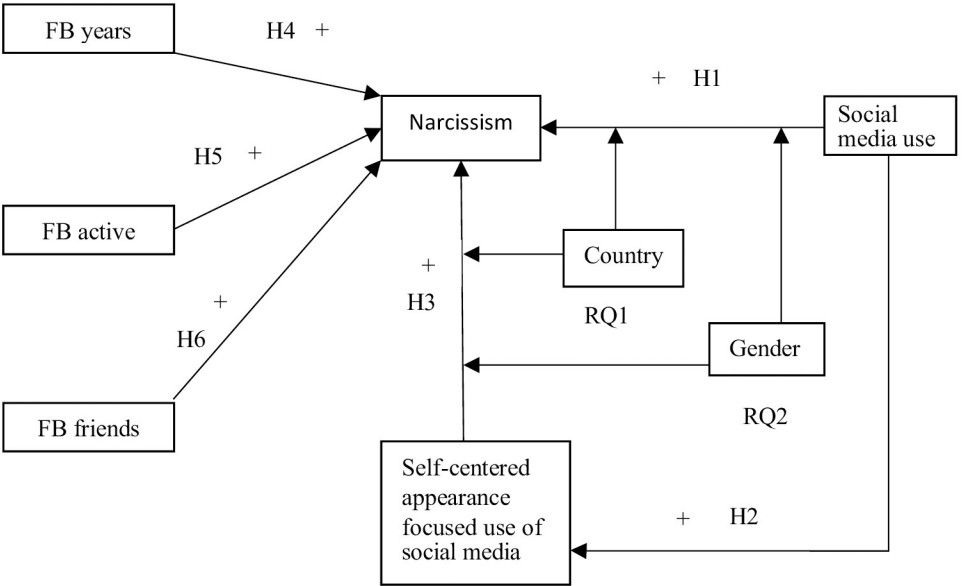

**Fig 1. Model of narcissistic social media usage.** Moderators are gender and culture.

was approved by the Ethics Committee of the University of Vienna (Austria) and the Social and Societal Ethics Committee (SMEC) of the University of Leuven (Belgium). Convenience sampling was used to recruit schools. Selection was based on a list of schools that was constructed by the researchers or was provided by the government in some of the countries. Schools were contacted through e-mail and/or by telephone until a substantial number of schools agreed to participate in order to guarantee a minimum of 300 participants in each. Early and middle adolescents (12 to 16 years old) were targeted in each country. Respectively seven, thirteen, five and four schools in Austria, Belgium, Spain, and South Korea agreed to participate.

We had previously sent to the school a written document in which parents had to give consent. This document was filled with the name of the participant and the name of one of the parents and manually signed by this parent. These documents were collected before the start of the data collection to make sure that no participant with no written parent consent would fill the questionnaire. Before participating in the study, adolescents were also asked to give active consent and were informed about the study goals.

Researchers visited the participating schools in each country with paper-and-pencil questionnaires and ensured anonymity and confidentiality during participation in the study. Depending on the customary habits of rewarding adolescent participants in each country, adolescents received an individual reward card (South Korea, 4$) or were entered in a lottery to win a reward card (i.e., Austria, Belgium, and Spain).

A total of 1,983 adolescents participated in our study. More precisely, 368 Austrian (18.6%), 675 Belgian (34%), 564 Spanish (28.4%) and 376 South Korean (19%) adolescents participated. The mean age was 14.41 years ($SD$ = 1.08), and 50.3% were boys. Adolescents mostly reported to have higher educated families as the majority of the sample (76.9%) reported that their mother and/or father had a higher educational degree. A MANOVA analysis revealed significant country differences regarding age, gender, and educational level of the parents, $V$ = .372, $F(9, 4602)$ = 72.29, $p < .001$, $\eta p^2$ = .124. Follow-up analyses indicated differences occurred for gender, $F(3, 1534)$ = 4.25, $p < .01$, educational level, $F(3, 1534)$ = 17.30, $p < .001$, and age $F(3, 1534)$ = 260.07, $p < .001$. adolescents from lower educated families participated in Austria ($M$ = 3.70, $SD$ = 0.90) as compared to Belgium ($M$ = 4.06, $SD$ = 1.06), Spain ($M$ = 4.18, $SD$ = 0.94), and South Korea ($M$ = 4.17, $SD$ = 0.91). Lastly, Belgian adolescents ($M$ = 13.74, $SD$ = 0.65) appeared to be younger than Austrian ($M$ = 15.15, $SD$ = 1.25), Spanish ($M$ = 15.09, $SD$ = 0.85), and South Korean ($M$ = 14.03, $SD$ = 0.82) adolescents.

## Measures

The questionnaire was originally designed in English. A backward translation- method was used to translate the questionnaire in all countries except for Spain. In Spain, two researchers translated the questionnaire through a two-step procedure. In a first step, each of the researchers translated half of the questionnaire. In a second step, they reviewed the translated questionnaire in order to further standardize terms. Special attention was also given in each country to align the wording of the scales to adolescents' daily living environment.

**Socio-demographic correlates.** Country (1 = *Austria*, 2 = *Belgium*, 3 = *Spain*, 4 = *South Korea*), gender (1 = *boy*, 2 = *girl*), educational degree of the mother and the father (1 = *no degree*, 2 = *elementary school degree*, 3 = *high school*, 4 = *professional bachelor*, 5 = *academic degree*), and age (birth year, recoded into age in years, $M$ = 14.41, $SD$ = 1.08) were requested.

**Social media use (SNS use).** Respondents indicated how often they used social media in general on a daily basis using a 7-point Likert scale ranging from "Never use it" to "More than 3 hours."

**Facebook and Instagram use.** Participants reported how much time they spent per day on Facebook or Instagram on a scale ranging from 1 (= never use it to less than 10 minutes) to

7 (= more than 6 hours). Respondents also answered questions about their active and passive Facebook/Instagram activities using a 5-point Likert scale ranging from "Never" to "Always" [69] to thirteen items. Eight items questioned respondents' involvement in active Facebook activities, such as "post comments on Facebook" and "update Facebook status." A principal components analysis indicated these items loaded on one factor (eigenvalue = 5.912, explained variance = 73.93%, alpha = .95). The items were averaged into one variable ($M$ = 1.92, $SD$ = 1.02). Five items asked about respondents' involvement in passive Facebook activities, such as "read comments on Facebook" and "check news feeds." A principal components analysis indicated these items loaded on one factor (eigenvalue = 3.79, explained variance = 75.80%, alpha = .92). The items were averaged into one variable ($M$ = 2.08, $SD$ = 1.15).

**Self-representation (Self-centered appearance presentation, SCA).** This concept operationalizes the self-centered appearance focused behavior with two self-reported scales. First, respondents used a 5-point Likert scale (1, *"almost never"*, 2, *"rarely"*, 3, *"sometimes"*, 4, *"often"*, 5, *"almost always"*) on statements about appearance in SNS (attractiveness): *"you take a selfie and post it on social networking sites"*, *"you use techniques to make you look better in pictures you post in social networking sites (cropping parts of yourself, using filters, using Photoshop or editing software)"*, *"you select the most attractive pictures of yourself to post on social media"*, *"you remove a tag identifying you on a photograph posted by another user"*, and the open question *"Could you explain why you usually remove it?"*) [32, 40, 70]. Second, the Physical Appearance Comparison Scale [71] is employed to measure self-representation in Facebook and Instagram. This scale presents a 5-point Likert scale (1, "never", 2, "seldom", 3, "sometimes", 4, "often", 5, "always"), on the following statements: *"When using Facebook/Instagram, I compare my physical appearance to the physical appearance of other"*, *"The best way for a person to know if they are overweight or underweight is to compare their figure to the figure of others on Facebook/Instagram"*, *"When using Facebook/Instagram, I compare how I am dressed to how other people are dressed"*, *"Comparing how you look to how others on Facebook/Instagram look is a bad way to determine if you are attractive or unattractive"*, and *"When using Facebook/Instagram, I sometimes compare my figure to the figures of other people"*.

**Narcissism.** Narcissism is measured using the Narcissistic Personality Questionnaire for Children-Revised (NPQC-R) [4] and the NPI-13 scale of narcissism [22]. The NPQC-R is a 12-item Likert scale ranges from 1 *"Not at all like you"* to 5 *"Completely like you"*, and respondents had to assess the narcissistic indicators *"I always know what I am doing"*, *"I am going to be a great person"*, *"I was born a good leader"*, *"I am really a special person"*, *"I think my body looks good"*, *"I think I am a great person"*, *"If I ruled the world it would be a better place"*, *"I am good at getting people to do things my way"*, *"It is easy for me to control other people"*, *"I would do almost anything if you dared me"*, *"I have the capacity to persuade people in believing anything I want them to"*, and *"When I am supposed to be punished, I can usually talk my way out of it"*. The NPI-13 scale consists of 13-items with two dichotomous statements as choices each. These statements inquire about attitude and behavior regarding authority, respect, power over others, showing off or body appreciation. Those are a few examples: *"I find it easy to manipulate people"* vs *"I don't like it when I find myself manipulating people"*, *"When people compliment me I get embarrassed"* vs *"I know that I am a good person because everybody keeps telling me so"*, *"I like having authority over other people"* vs *"I don't mind following orders"*, and so on.

## Results

The dataset used to extract results is deposited in the public repository http://Osf.io (URL: https://osf.io/f2vwg/?view_only=acfafa0d822542dba30a40d4f007c075). Before we dive into the test of hypothesis, we provide some preliminary descriptive statistics. The overall mean ($M$)

**Table 1. Preliminary descriptive statistics.**

| Variables | Mean | Mode | SD |
|---|---|---|---|
| NPQC-R | 3.05 | 3 | 0.728 |
| NPI-13 | 3.70 | 3 | 2.403 |
| SNS use | 2.75 | 2.71 | 1.085 |
| SCA | 1.94 | 1 | 0.835 |
| Active FB use | 1.95 | 1 | 1.041 |
| FB years | 3.25 | 3 | 1.585 |
| FB friends | | 3 | 2.319 |

for NPQC-R was 3.05 (Standard Deviation, henceforth $SD$ = .728) (of which 911 above the mean and 946 below) and the NPI-13 mean was 3.70 ($SD$ = 2.403). Social media use (SNS use) mean was 2.75 ($SD$ = 1.085), self-centered appearance focused use of social media (SCA) was 1.94 ($SD$ = .835) and active Facebook use (the average of the 13 items measure Facebook and Instagram active use) was 1.95 ($SD$ = 1.041). Years of Facebook use averaged 3.25 years ($SD$ = 1.585) and the most reported Facebook friends span was 101–200 friends (span 3), although the mean was higher (202–300 friends; $SD$ = 2.319) (see Table 1; The abbreviations in this table mean the following: NPQC-R: Narcissistic personality questionnaire for Children-Revised Likert 12 items; NPI- 13: Narcissistic Index; SNS Use: Social Networks Use -hours spent; SCA: self-centered appearance focused use of social media; Active FB use: Active Facebook use; FB Years: years of Facebook use; FB Friends: number of friends in Facebook or Instagram).

Measurement invariance for the NPI-13 and NPQC-R scale was examined with a correlation test. Both variables, NPQC-R and NPI-13 correlate ($r$ (1686) = .505, $p$ < .001) so it can be assumed that they measured narcissism in a similar way. Nevertheless, we report the two separately to add clarity to findings. Since NPQC-R offered more consistent results (and because it is based on Likert scales), it was used for splitting high vs low narcissism in participants (using the Mean as a high/low threshold as informed next).

In response to H1 (H1: A higher social networks use will be related positively with narcissism), social media use (SNS use) correlated positively with narcissism in both NPQC-R ($r$ (1716) = 0.0744, $p$ = .002) and NPI-13 ($r$ (1654) = 0.203, $p$ < .001), meaning that a higher usage of social networks is related with higher narcissism, in all four countries, and for males and females. The gender distribution was part of the research question, so this partially contributes to respond to it (RQ: What are the country and gender variations in the relationship between social media use (SNS use) and narcissism (H1), SNS use and self-centered appearance focused use (SCA) of SNS (H2), and SCA and narcissism (H3)?).

So, within this research question, males reported lower overall SNS use ($M$ = 2.66, *1.07*) than females ($M$ = 2.84, $SD$ = 1.1), and the correlation between SNS use and narcissism was higher for males than for females (males NPQC-R $r$ (841) = .0251, $p$ < .001; NPI-13 $r$ (803) = 0.2, $p$ < .001; females NPQC-R $r$ (865) = .159, $p$ < .001; NPI-13 $r$ (839) = .0399, $p$ < .001). As can be seen in Table 2, male adolescents scoring high in narcissism had a mean of 2.75 in social media use (SNS use) whereas those scoring low in narcissism had a mean of 2.55; while in the case of female adolescents, the difference of SNS use between those scoring high and those scoring low in narcissism is smaller (2.89 and 2.79 respectively). Narcissism is calculated here as a dichotomic variable between those that are over the mean average (>3.05) (high narcissism) and those that are under the mean average (>3.05) (low narcissism) in the NPQC-R index.

**Table 2. Social media use (SNS use) by narcissism and gender.**

| Narcissism | Gender | SNS use |
|---|---|---|
| High | Male | 2.75 |
| | Female | 2.89 |
| Low | Male | 2.55 |
| | Female | 2.79 |

The threshold to divide segments between high and low narcissism is the mean average of this variable, that is, higher or lower than 3.05 (as also reported in Fig 2).

Narcissism was more significant ($p < .001$) than gender ($p = .002$) when comparing the correlation between social media use (SNS use) and narcissism (using the threshold for NPQC-R $M \geq 3.05$, since NPI-13 showed no significant differences by gender). As can be seen in Fig 2, although SNS use for females was higher, males showed a stronger association between high SNS use and high narcissism ($t (1716) = 2.41$, $p = .016$).

Still within the scope of H1 and RQ (relationship between social media use and narcissism), we now turn to see how this relationship can be moderated by country of origin. All countries except South Korea's NPI-13' scale of narcissism reported significant correlations between narcissism and social media use (SNS use) (all $p$-values $<.005$). Spanish teenagers show the

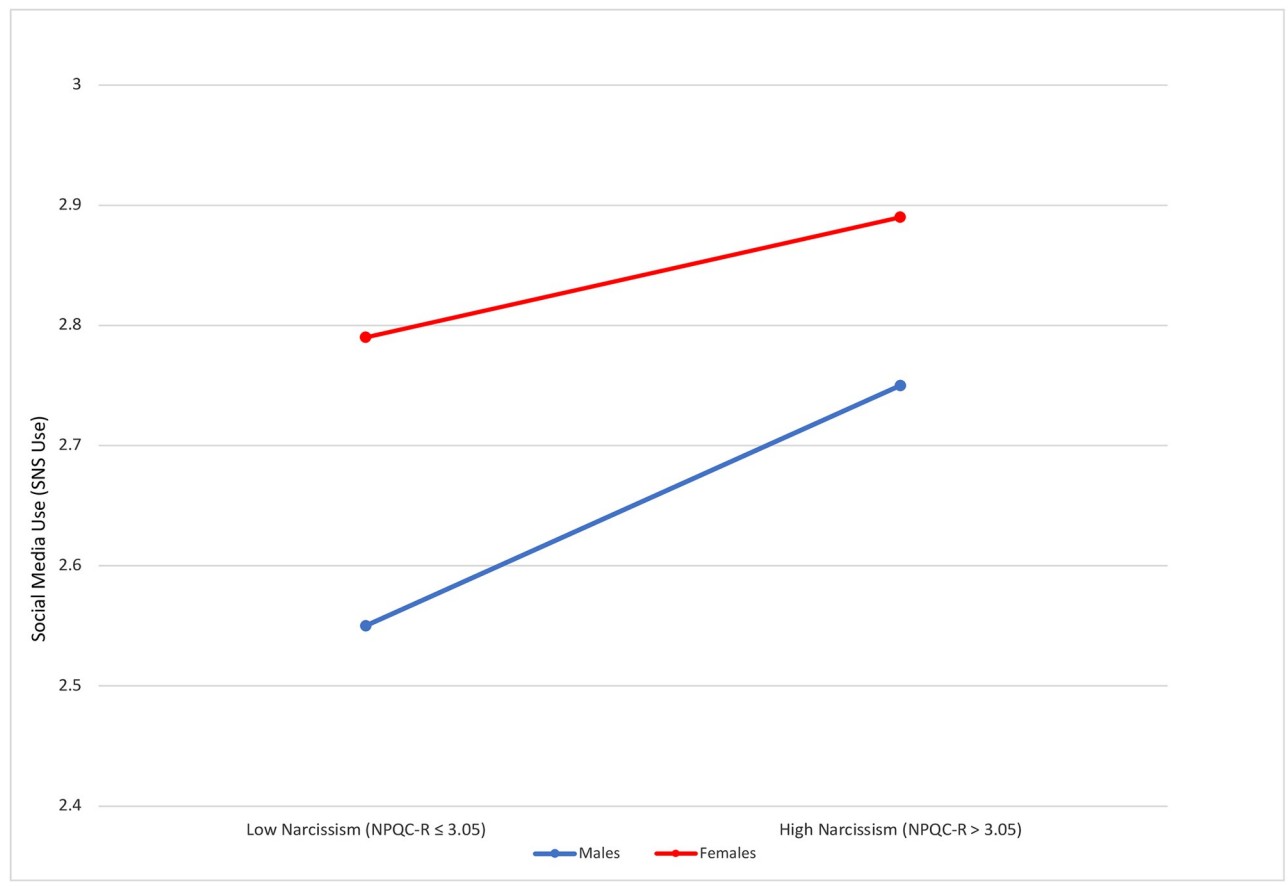

**Fig 2. Social media use mean (SNS use) by gender and narcissism.**

Table 3. Social media use (SNS use) and narcissism (NPQC-R and NPI-13)*.

| Country | SNS use | NPQC-R | NPI-13 |
|---|---|---|---|
| Austria * | 2.72 | 3.04 | 4 |
| Belgium * | 2.71 | 2.82 | 3.11 |
| South Korea | 2.41 | 3.40 | 3.93 |
| Spain * | 3.07 | 3.07 | 4.13 |

* Correlation between SNS use and Narcissism (NPQC-R or NPI-13) p < 0.05, except NPQC-R South Korea.

strongest correlation between social media use and both variables of narcissism (NPQC-R, r (493) = .159, $p < .001$; NPI-13, r (462) = .2, $p < .001$) and, among the significant correlations, they reported the highest values of social media use (SNS use) (3.07) (Table 3). Austria and Belgium show correlation between SNS use and narcissism and a similar social media use (2.72 and 2.71) (NPQC-R, $r$ (570) = .0947, $p = .023$; NPI-13, $r$ (566) = .224, $p < .001$ for Belgium; and NPQC-R, $r$ (287) = .114, $p < .053$; NPI-13, $r$ (261) = .214, $p < .001$ for the Austrians) while South Korea had a lower social media use (2.41) and no significant correlation between SNS use and narcissism (NPQC-R).

Furthermore, we narrow the focus by comparing the differences between countries, narcissism, gender, and social media use (SNS use) (all correlation $p$-values under .05, except South Korea). Since the high versus low narcissism is calculated using the NPQC-R index to add consistency and rigor, and South Korean adolescents showed no correlation between SNS use and NPQC-R, the ANOVA model does not consider this country (Table 4).

Now, data from Austrian and Belgian male adolescents clearly show the reported correlation between SNS use and narcissism, Spanish adolescents showing the highest correlation for both female and male adolescents. As seen graphically in Fig 3 and numerically in Table 4, Spanish males and females are the ones that more clearly show a connection between using social media more intensively (3.12 and 3.31 respectively in the scale) and reporting higher levels of narcissism. Similarly, Austrian and Belgian males show the same pattern with almost the same values of social media use (2.83 and 2.82), whereas females reporting high narcissism in these two countries score 2.68 and 2.75 of social media use respectively.

Next, in response to H2 (H2: A higher social media use will be related with higher self-centered appearance focused use of social media), there is a positive correlation (Fig 4) between

Table 4. Social media use (SNS use) by country, gender, and level of narcissism.

| Country | Gender | Narcissism | SNS use |
|---|---|---|---|
| Austria** | Male | High | 2.83 |
| | | Low | 2.63 |
| | Female | High | 2.68 |
| | | Low | 2.65 |
| Belgium** | Male | High | 2.82 |
| | | Low | 2.52 |
| | Female | High | 2.75 |
| | | Low | 2.76 |
| Spain** | Male | High | 3.12 |
| | | Low | 2.75 |
| | Female | High | 3.31 |
| | | Low | 3.04 |

** ANOVA overall model p<.001 (all countries)

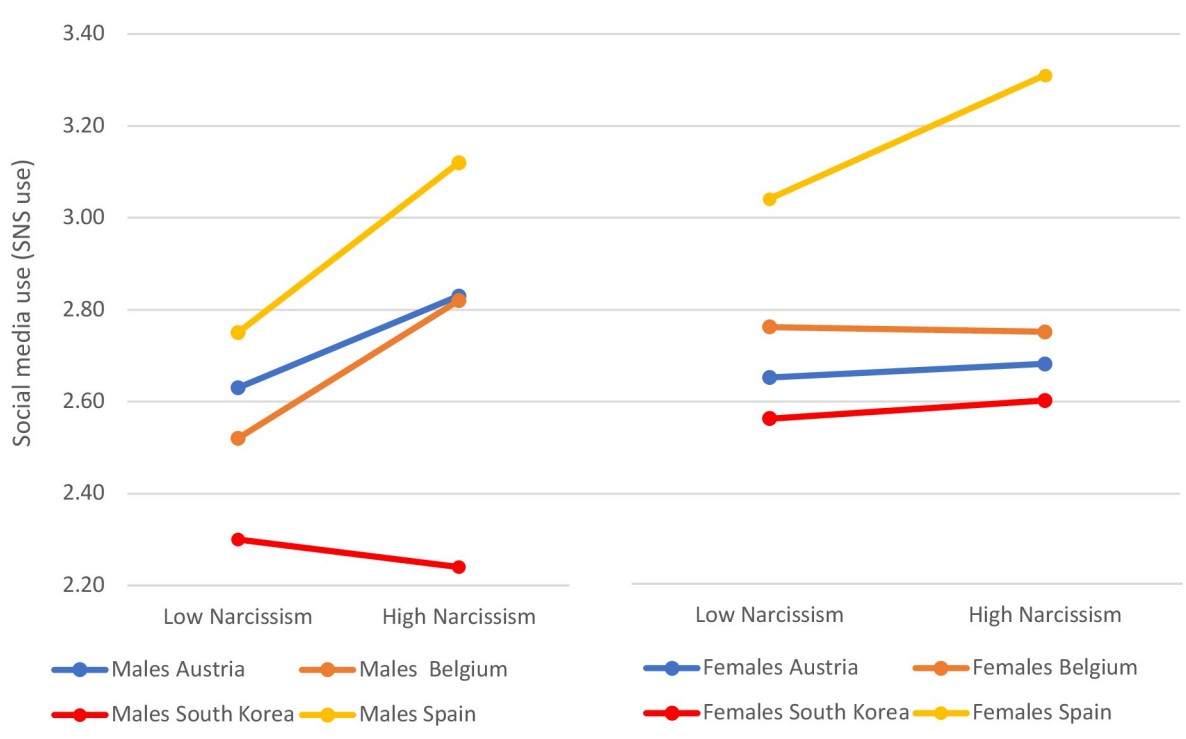

**Fig 3. Social media use (SNS use) by gender, narcissism, and country.**

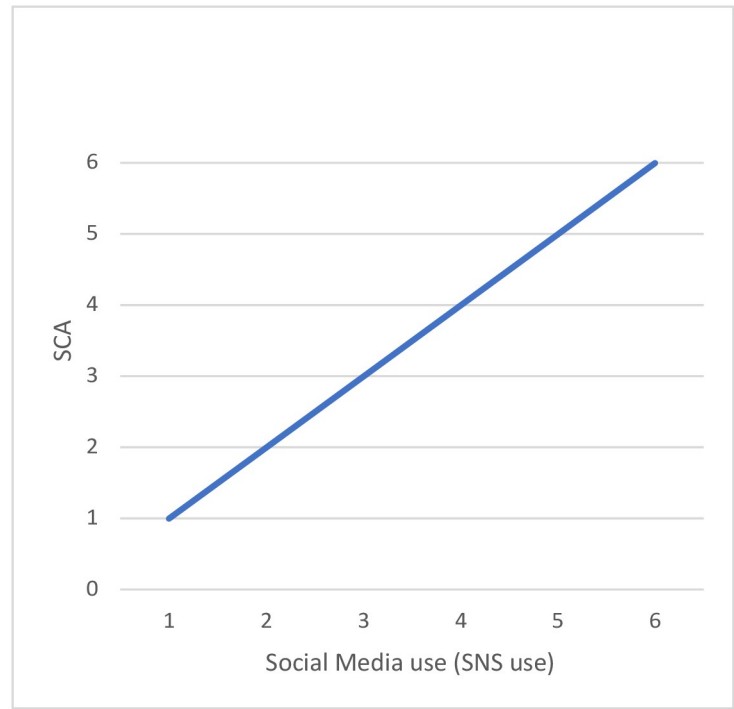

**Fig 4. Self-centered appearance focused use of social media (SCA) correlation with social media use (SNS use).**

**Table 5. Self-centered appearance focused use of social media (SCA) and social media (SNS use) use by gender.**

|         | Gender    | Mean | SD    |
|---------|-----------|------|-------|
| SNS Use | Male**    | 2.66 | 1.067 |
|         | Female**  | 2.84 | 1.099 |
| SCA     | Male**    | 1.72 | 0.756 |
|         | Female**  | 2.17 | 0.848 |

** Correlation between SCA and SNS p<.001 per gender

SNS use and self-centered appearance focused use of social media (SCA) ($r$ (1702) = .411, $p <$ .001), meaning that someone makes use social media, the more prone he or she is to focus on one self's appearance.

Further, the SCA variable (self-centered appearance focused use of social media) is explored here as a high-low condition too (the threshold being $M$ = 1.94), showing that a high social media use entails a high SCA use ($M$ = 3.13, $SD$ = 1.075 versus $M$ = 2.4, $SD$ = 0.059). Differences were confirmed with a t-test ($t$ (1702) = 14.8, $p <$ .001).

H2 is also confirmed for males and females separately (RQ) (females: $r$ (845) = .401, $p<$.001; males: $r$(848) = .402, $p <$ .001), both groups showing similar results ($p$ = .407). In both, more social media use (SNS use) was related with more SCA. Males ($M$ = 2.66, $SD$ = 1.168) used social media more intensively than females ($M$ = 2.84, $SD$ = 1.223) but reported a less self-centered use (SCA) (males, $M$ = 1.72, $SD$ = 0.721; females, $M$ = 2.17, $SD$ = 0.775) (Table 5).

Analyzing the self-centered appearance focused use of social media (SCA) as a high versus low variable, we can visually see the similarities between a high versus low self-centered

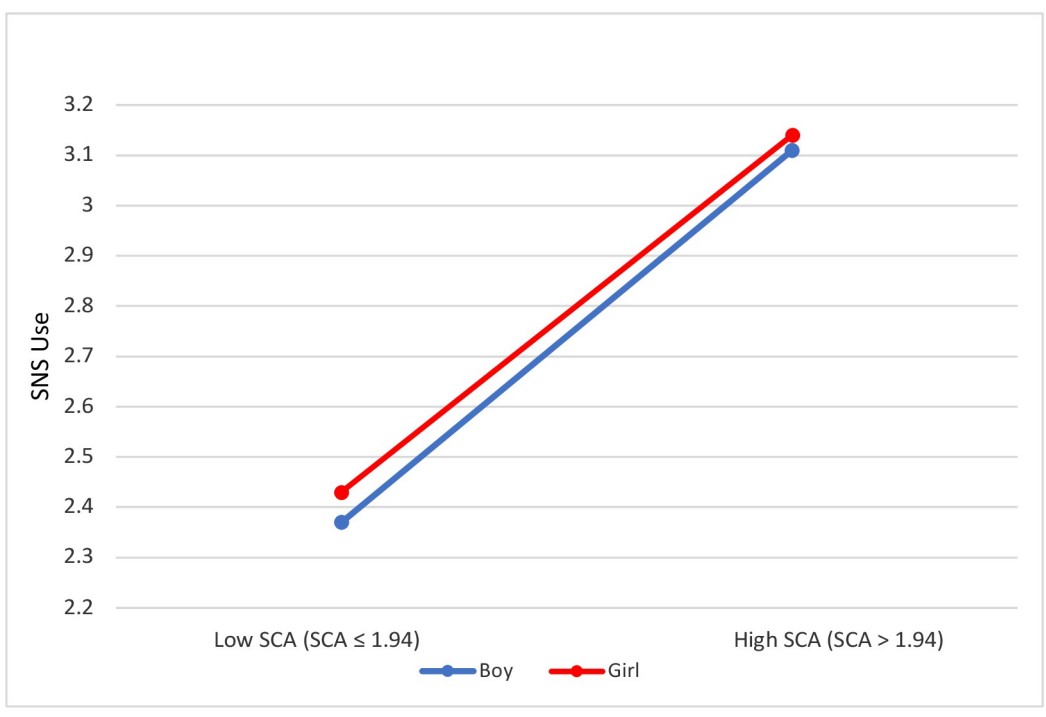

**Fig 5. Social media use (SNS use) in girls and boys works similarly between high/low self-centered appearance focused use of social media (SCA).**

**Table 6. Social media use (SNS use) and self-centered appearance focused use of social media (SCA) for each country.**

| Country | SNS use | SCA |
| --- | --- | --- |
| Austria* | 2.72 | 2.24 |
| Belgium* | 2.71 | 1.91 |
| South-Korea* | 2.41 | 1.70 |
| Spain* | 3.07 | 1.98 |

* Correlation between SCA and SNS p<.05

appearance focused use of social media (SCA) by males ($M$ = 3.11, $SD$ = 1.039 and 2.37, $SD$ = 0.954) and females ($M$ = 3.14, $SD$ = 1.101 and 2.43, $SD$ = 0.97) (see Fig 5).

In addition, no country differences were found, hence all countries show positive correlations between SNS use and SCA (all $p$-values under 0.05). Spanish adolescents reported the higher SNS use, but medium SCA (1.98); then, Austrian adolescents reported a 2.72 of SNS use and a 2.24 of SCA; Belgian adolescents a 2.71 and 191 respectively, and South Korean adolescents 2.41 and 1.70 (see Table 6).

When splitting SCA into high versus low categories (all $p$-values under 0.001), we observed a positive correlation for each country (Fig 6 and Table 7), stronger for Belgium (3.11 of social media use for those with high SCA as opposed to 2.31 of social media use for those with low SCA) and South-Korea (3.35 and 2.75 respectively) if compared with Austria (2.96 and 2.34) and Spain (2.91 and 2.17).

In response to H3 (H3: Self-centered appearance focused use of social media correlates positively with narcissism), a positive and significant correlation was found between the self-

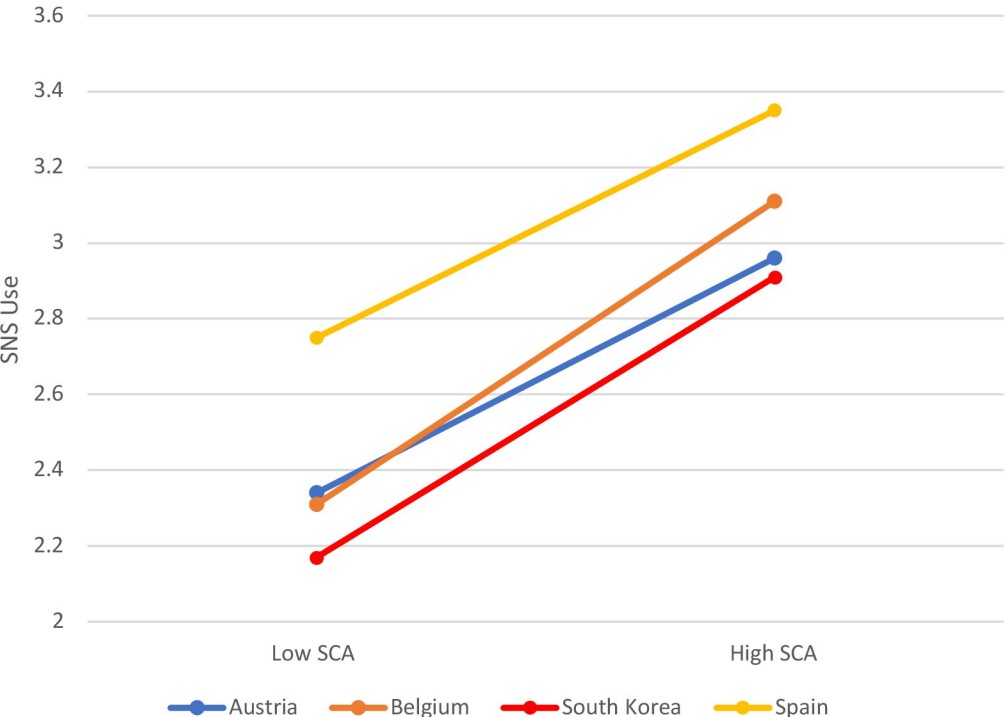

**Fig 6. Social media use (SNS use) in low and high self-centered appearance focused use of social media (SCA).**

**Table 7. Social media use (SNS use) by country, and level of narcissism.**

| Country** | SCA** | SNS use |
|---|---|---|
| Austria** | High | 2.96 |
| | Low | 2.34 |
| Belgium** | High | 3.11 |
| | Low | 2.31 |
| South-Korea** | High | 3.35 |
| | Low | 2.75 |
| Spain** | High | 2.91 |
| | Low | 2.17 |

** Correlation between SCA and SNS use p<.001 per country

centered appearance focused use of social media (SCA) and NPQC-R ($r$ (1728) = 0.094, p<.001), and NPI-13 ($r$(1669) = 0.263, $p$<.001). Dichotomizing narcissism as a high-low measure (NPQC-R threshold M = 3.05) further reinforces the positive relation with SCA (t(1728) = 3.30, p = .001; High $M$ = 2.01, Low $M$ = 1.88).

Gender was a significant variable on narcissism NPQC-R index, but not significant for NPI-13 (NPQC-R $t$(1847) = 0.094, $p$<.001, NPI-13 $t$(1755) = 1.86, $p$ = .0.63). The ANOVA tests were significant for high and low narcissism and for gender on SCA (all tests $p$<.001). No interactive effects were found.

As observed, regardless of a higher social media use by females in general, the correlation of SNS use and SCA is very similar for males and females (an increase from 2.08 of social media use for females with low SCA to 2.28 for females with high SCA; whereas males' social media use increases from 1.63 to 1.79) (see Table 8 and Fig 7).

The correlation between narcissism and SCA was positive and significant for all countries too (all p-values below 0.05). Overall, Austria reported the highest levels of SCA ($M$ = 2 and 2.63) and South Korea the lowest ($M$ = 1.38 and 2.21) for males and females reporting high narcissism. The cases of Belgium and Spain were similar since narcissist males and females scored 1.91 and 2.14 respectively in the case of Belgium, and 1.95 and 2.22 in the case of Spain (Table 9 and Fig 8).

We turn now to hypothesis 4, 5 and 6 (H4, H5, and H6), which provide with insightful data on Facebook (FB) use and narcissism. H4 is confirmed since an active use of FB correlates with narcissism in both indexes: NPI-13 (r (1646) = .0908, p < .001) and with NPQC-R index (r (1719) = .768, p = .001). And the same for H5 since the number of FB friends also correlates with narcissism in both indexes: NPI-13 (r (1017) = .164, p < .001) and NPQC-R (r (1037) = .148, $p$<.001); and for H6: the number of years in FB correlates with narcissism in both indexes, mainly NPI-13 (r (1005) = 1.113, p < .001), and, to a lesser extent, NPQC-R (r (1020) = .0964, $p$ = .002).

**Table 8. Self-centered appearance focused use of social media (SCA) as associated with levels of narcissism and gender.**

| Gender | SCA | SNS use |
|---|---|---|
| Males* | High | 1.79 |
| | Low | 1.63 |
| Females* | High | 2.28 |
| | Low | 2.08 |

** Overall model test Gender*SCA on SNS use p<.001

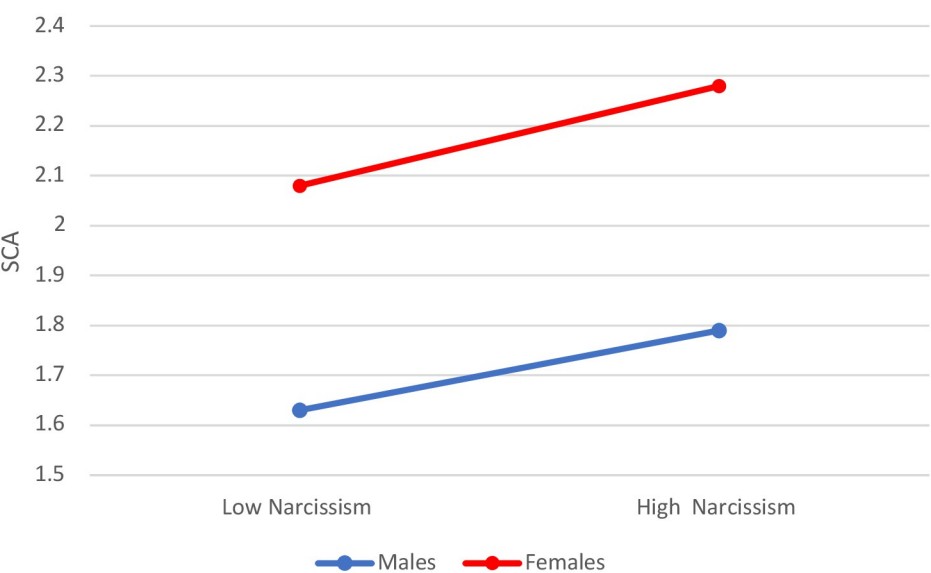

**Fig 7. Self-centered appearance focused use of social media (SCA) across low/high narcissism males and females.**

Finally, mediation analysis was performed to assess the mediating role of the self-centered appearance focused use of social media (SCA) between social media use (SNS use) and narcissism (Bootstrap = 1000 in both indexes). For NPQC-R, both the total effect ($\beta$ = .0468, z = 2.5, $p$ = .013) and the indirect effect ($\beta$ = .0179, z = 2.22, $p$ = .027) were significant, but the direct effect was not ($\beta$ = .0289, z = 1.45, $p$ = .148). For NPI-13, mediation of NCA was confirmed, results revealed a positive total effect ($\beta$ = .458, z = 7.92, $p <$ .001), indirect effect ($\beta$ = .191, z = 6.82, $p <$ .001) and direct effect accounting SCA ($\beta$ = .267, z = 4.47, $p <$ .001). Indirect effect of SCA explained 41.6% of the mediation for NPI-13 and 38.2% of the moderation for NPQC-R (see Fig 9).

**Table 9. Self-centered appearance focused use of social media (SCA) by countries, gender, and low/high narcissism.**

| Country | Gender | Narcissism | SCA | SD |
|---|---|---|---|---|
| Austria | Male | High | 2.00 | 1.004 |
| | | Low | 1.86 | 0.749 |
| | Females | High | 2.63 | 0.886 |
| | | Low | 2.33 | 0.910 |
| Belgium | Male | High | 1.91 | 0.769 |
| | | Low | 1.71 | 0.678 |
| | Females | High | 2.14 | 0.790 |
| | | Low | 2.02 | 0.769 |
| South Korea | Male | High | 1.38 | 0.701 |
| | | Low | 1.16 | 0.430 |
| | Females | High | 2.21 | 0.943 |
| | | Low | 1.96 | 0.873 |
| Spain | Male | High | 1.95 | 0.724 |
| | | Low | 1.60 | 0.607 |
| | Females | High | 2.22 | 0.878 |
| | | Low | 2.07 | 0.671 |

** Overall model Country*Gender*Narcissism on SCA p<.001

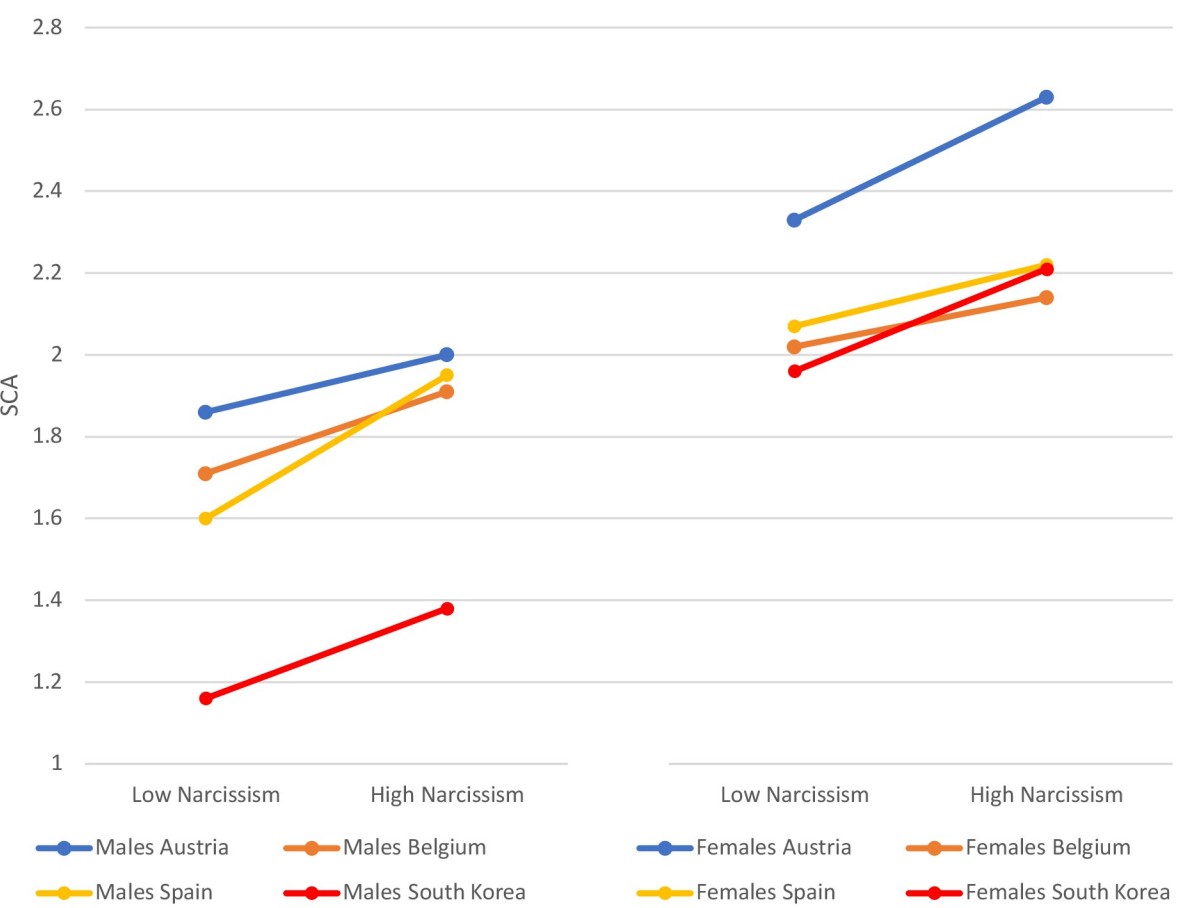

**Fig 8. Self-centered appearance focused use of social media (SCA) among low/high narcissism boys and girls by countries.**

In sum, although all independent correlations had been confirmed (H1, H2 and H3), SCA mediation on NPQC-R was not conclusive. NPI-13 showed more consistently a mediation of SCA on the SNS and the narcissism link, while NPQC-R showed that direct effect including SCA was not significant (see Fig 10).

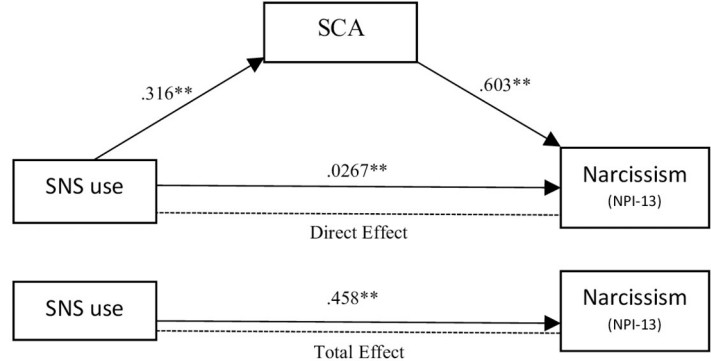

**Fig 9. Mediation and total effect of the self-centered appearance focused use of social media (SCA) on the link between SNS and narcissism NPI-13.**

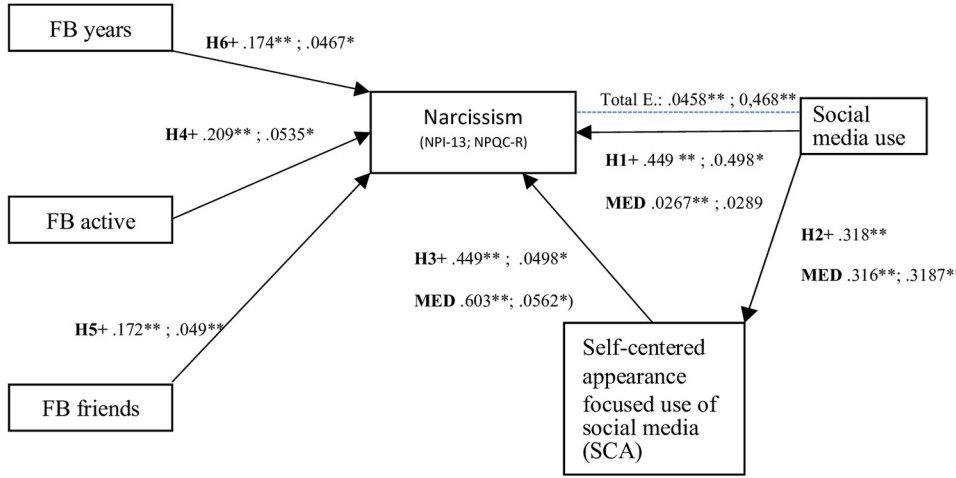

Estimates for correlations or mediation (MED) estimates are (NPI-13 ; NPQC-R).
* p-value <.05
** p-value <.001

**Fig 10. Model of narcissistic social media usage.**

## Discussion

The correlation between social media use (SNS use) and narcissism is positive for all countries and for males and females. When making a high/low distribution of narcissism, there were no correlation for South Korean males and very weak correlation for females. Mainly, the three European countries [11] are in line with previous findings on, first, the fact that female adolescents use social media more intensively regardless of reporting low or high narcissism, and second, male adolescents show a greater association between a high usage of social media and narcissism [65]. This is particularly clear for Spanish males as opposed to the case of South Korean adolescents. An analogous cultural difference had been found previously between Spanish adolescents and Thai adolescents [61].

The second finding is the relevant association between social media use and a self-centered appearance focused usage of social media (SCA). For both males and females from the four countries, the more adolescents use social media, the more likely these adolescents will make a self-centered appearance focused use of them. Being the case that the SCA (self-centered appearance focused use of social media) can be conceived as a narcissistic type of SNS usage, the confirmation of this hypothesis adds relevance to the connection between using social media and narcissism (H1) and the further connection between SCA and narcissism (H3). Again, a SCA use of social media is connected with narcissism, but the connection is greater for Austrian females and weaker for South Korean males and Belgian females.

In addition, SCA has showed to be a moderating factor of the link between social networks use (SNS) and narcissism (especially on the NPI-13 index). According to this regression analysis it seems plausible that the increased use of social media increases narcissistic personality traits, or else, that narcissistic profile is more prone to have a SCA use of SNS. Moreover, a self-interested use of social networks, increases with both narcissism and self-centered use. In this line of thought, this study also presents more specific data connecting the use of FB, the number of friends and the years of use of this social media platform with both measures of narcissism. These correlations are demonstrative on the factual connection between these specific

phenomena and the reported narcissism, especially the extended use of Facebook throughout the years.

Furtherly, although the correlation between social media use, the self-centered appearance focused use of social media (SCA) and narcissism was not significantly different for females and males, we should remark that females reported higher levels of social media use and SCA than males, but lower levels of narcissism and, in fact, females show weaker correlation in the three hypothesis -which can be clearly seen in the low versus high division of narcissism. This aligns with Errasti et al.'s study [61] for the Spanish sample in the case of Facebook. Consistently, males score higher in narcissism [15, 17] and show a stronger correlation in the hypotheses 1, 2 and 3.

Thus, the self-centered appearance focused use of social media (SCA) emerges here as a relevant mechanism to furtherly pose directionality in the connections of social media use (SNS use), SCA and narcissism. SNS may be used in a non-narcissistic way and thus have no direct relationship with narcissism [9], but in the light of our findings, SNS can also be used in a narcissistic way, presumably by narcissists, so that using SNS is connected with a SCA type of use and, ultimately, narcissism, regardless of gender and country of origin. In sum, in line with Buffardi and Campbell's study [31] and Slater's general theory on the reinforcing spirals effect (2007), social media can permit a narcissistic self-regulation. This would explain previous contradictory findings on the directionality of effects between social media usage and narcissism [32, 33], being the case that some of those previous studies were not supported by a strong media theory [39].

The contribution of the study is thus on the media theorization for this phenomenon. As put by Slater, 'the cognitive or behavioral outcomes of media use also influence media use, particularly when the cognitions or behaviors are related to personal or social identity' [37: 283]. This approach has been incorporated to our topic of interest here in the form of specific self-centered behaviors in social media as mere displayers of narcissism but also as implicit catalyzers -reinforcers following Slater- of narcissism [9] on the basis of, again, correlations between the two ends: narcissism and social media use.

In sum, the prevalence of narcissism as a personality trait triggered by a variety of socialization inputs, many of which are mediated, is shown here with a cross-cultural study on a large and homogenous sample of adolescents [72].

## Conclusions

The study presented here consisted in a survey conducted physically in four different countries with a cross culturally and gender balanced sample of 1,983 adolescents from 12 to 16. The objective was to explore how the intensity of social media use (SNS use) is associated with narcissism through a self-centered appearance focused use of SNS; and to control country and gender variations. We posed a model of correlations that resulted in a mediation model, showing that a self-centered appearance focused use of social media (SCA) moderates the association between SNS use and narcissism. Since SCA is defined in the study as narcissistic behavior in SNS, we draw on the reinforcing spirals theory [37] to posit the integration of SNS in the socialization process, which commonly leads to a narcissistic personality and narcissistic behavior both online and offline.

The approach brought by this study is not without limitations. Clearly, a more accurate statistical model such as the structural equation modelling on the role of variables, and a more filtered and homogenous sample, could provide evidence and exact power effect to the causality of specific narcissistic behaviors in SNS -e.g., exhibitionism- on personality formation. Besides, ethnographic techniques should also be part of the equation. This line of research needs to

make progress due to the increasing range of social media tools and the increasing relevance and availability of those at all levels -personal, social, and professional, with several implications for intimate relationships, leisure, sports, or family, to name just a few.

## Supporting information

**S1 Data. Dataset Intercultural Study Project (ISP).**
(XLSX)

## Author Contributions

**Conceptualization:** Lluís Mas Manchón.

**Data curation:** Lluís Mas Manchón, David Badajoz Dávila.

**Formal analysis:** Lluís Mas Manchón, David Badajoz Dávila.

**Funding acquisition:** Lluís Mas Manchón.

**Investigation:** Lluís Mas Manchón.

**Methodology:** Lluís Mas Manchón, David Badajoz Dávila.

**Project administration:** Lluís Mas Manchón.

**Resources:** Lluís Mas Manchón.

**Software:** Lluís Mas Manchón, David Badajoz Dávila.

**Supervision:** Lluís Mas Manchón.

**Validation:** Lluís Mas Manchón, David Badajoz Dávila.

**Visualization:** Lluís Mas Manchón, David Badajoz Dávila.

**Writing – original draft:** Lluís Mas Manchón.

**Writing – review & editing:** Lluís Mas Manchón, David Badajoz Dávila.

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
