## [Decision Letter · Decision Letter 0]

27 Apr 2022

PONE-D-21-34574The use of social media as a two-way mirror for narcissistic adolescents from Austria, Belgium, South-Korea, and SpainPLOS ONE

Dear Dr. Mas Manchón,

Thank you for submitting your manuscript to PLOS ONE. After careful consideration, we feel that it has merit but does not fully meet PLOS ONE’s publication criteria as it currently stands. Therefore, we invite you to submit a revised version of the manuscript that addresses the points raised during the review process.

Please carefully address the reviewers' concerns. Particular attention is needed on improving the description of the methodology and depth of the evaluation results and analysis.  The quality of figures and the overall presentation of the manuscript should be improved. Make sure to consider the PLOS ONE’s publication criteria in preparing your revision.

We look forward to receiving your revised manuscript.

Kind regards,

Rashid Mehmood, PhD

Academic Editor

PLOS ONE

Journal Requirements:

Reviewers' comments:

Reviewer's Responses to Questions

**Comments to the Author**

1. Is the manuscript technically sound, and do the data support the conclusions?

Reviewer #1: No

Reviewer #2: No

2. Has the statistical analysis been performed appropriately and rigorously? 

Reviewer #1: No

Reviewer #2: No

3. Have the authors made all data underlying the findings in their manuscript fully available?

Reviewer #1: Yes

Reviewer #2: No

4. Is the manuscript presented in an intelligible fashion and written in standard English?

Reviewer #1: Yes

Reviewer #2: No

5. Review Comments to the Author

Reviewer #1: This study concerned studying and understanding whether social networking sites - Facebook and Instagram- contribute to narcissistic behaviour . Therefore, the purpose of this study is to examine how intensities of SNS use are associated with narcissism through a self-centered appearance-focused SNS use, in addition to exploring whether cultural differences and gender play a role during preadolescence and early adolescence. The study involves a physical survey conducted in four different countries: Austria, Belgium, Spain, and South Korea with a cross-cultural and gender-balanced sample of 1,983 adolescents aged 12-16. Their methods to collect the data follow step by step approach, that starts from getting the agreement of the parents and then meeting the students to inform them about the study goals. Next, fill out the questionnaires physically. According to the authors, self-centered appearance-focused use of SNS (SCA) moderates the association between SNS use and narcissism. Based on their definition of SCA as narcissistic behaviour in social networks, they proposed the reinforcing spiral theory to explain the integration of SNS in the development of narcissistic behaviour, both online and offline. The paper is well written and follows a standard rational development, methodology, and analysis. The authors need to improve the presentation and discussion of the results.

Reviewer #2: The authors in this paper investigate how the intensity of SNS use is associated with narcissism through a self-centered appearance focused use of these SNS; and whether these associations are moderated or not by cultural differences of the country of origin in such a critical age of personality formation and (global) culturalization as the transition from pre-adolescence to adolescence. A correlation and mediation analysis were performed on cross-sectional survey among 1983 adolescents from Austria, South-Korea, Belgium, and Spain. They found that a self-centered appearance focused use of SNS (SCA) moderates the association between SNS use and narcissism, especially for males from the three European countries. It was also found that the years of use, number of friends and time spent in FB are associated with narcissism.

Comments:

- The authors should further explain the tables (e.g., what the columns represent, explain abbreviations, what does High and Low mean, what is threshold,

- The results are not properly explained. Some variables are not explained e.g. RQ, M. How they are obtained, what they represent. equations need to be provided

- Some values are presented in the text but are not provided in the table

- With respect to the Result.

o Further interpretation is needed.

o SD is not clear

o Line 378, what are r, p, and how they are obtained?

o Line 388, what is RQ?

o Line 389, what is M?

o Line 399, “(t (1716) = 2.41, p = .016).” are these numbers presented in a table??

o Table 1,

Not properly explained

It was not mentioned in the text.

What does first column represent? What is FB years?

Where are the values for Mode Column?

What is N column?

o In table 2,

why value of country is High and Low? What does it mean? On what basis they are considered Low and High??What is the threshold.

2nd column should be Gender (Mistake)

o Line 404, All countries except South Korea reported positive correlations for H1 (all p-values <.005 except South Korea’s NPQC-R) ? is this shown in the table?

o Line 406, “Spanish teenagers reported the highest values of SNS use (Table 3), highest narcissism levels.” It is not the highest narcissism levels because South Korea is higher.

o Table 3,

in its caption, they mentioned NPI-13 but is it not in the table

How p-values are obtained?

o Line 415.” all p-values under .05, except South Korea)” please explain.

o Line 445, “Table 6. Social media use (SNS use) and self-appearance focused behavior use of SNS”, in other table captions SNS is mentioned without the complete phrase. Consistency is required.

- Please explain Table 4

- Figures quality is not good.

- Figure 1 is not mentioned in the text.

- Fig 2, what is x-axis, what is y-axis?

- Figure 4, give full form of SCA should be added.

- For Figure 2,3, and 5, the threshold should be shown in the Figures.

6. PLOS authors have the option to publish the peer review history of their article (what does this mean?). If published, this will include your full peer review and any attached files.

Reviewer #1: No

Reviewer #2: No

---

## [Author Response · Author response to Decision Letter 0]

18 May 2022

We thank the reviewers for their relevant and useful comments. Reviewer #2 is addressed individually, with the reviewer’s comments followed by our answers.

Reviewer 2

The authors should further explain the tables (e.g., what the columns represent, explain abbreviations, what does High and Low mean, what is threshold, 

As explained next in each comment, some tables had wrong information or were not clear. Following the changes proposed by the reviewers, we have edited the tables, explained them in the text and have deleted two of the tables (7 and 10) so that the Results sections is much clearer now. We have provided the figures as editable PDFs files so that the quality is high. 

The results are not properly explained. Some variables are not explained e.g. RQ, M. How they are obtained, what they represent. equations need to be provided

RQ is the Research Question (we deleted this mention on the Results because it could be misleading). M’s are means or averages calculated for each variable. We have reviewed the text in order to further specify to what variables refer each M value. In some cases, we have added the specific equation (correlation, student-t or ANOVA) that was used, just before the result in APA format, for clarification. As mentioned, Results have now a clearer structure and the reference to the tables information is more explicit so that they are much easier to follow. 

Some values are presented in the text but are not provided in the table 

NPI-13 was mentioned in the text but it was not in table 3. It has been added. We have also found that the mode was not informed in table 1, so it has also been added too. We have reviewed all the connections between tables and text. 

With respect to the Result: Further interpretation is needed; 

SD is not clear; Line 378, what are r, p, and how they are obtained?; 

We have clarified these data and added that the test is a correlation test. We have made clear that the r is the correlation coefficient, this is, the strength and direction of the association of variables, and the p is how significant/relevant is this direction, accepted if under .05. 

Line 388, what is RQ?; Line 389, what is M?; Line 399, “(t (1716) = 2.41, p = .016).” are these numbers presented in a table?? 

We deleted the RQ in this line since it is a mistake. M is “Mean” (or average) as per APA abbreviation. With regard to t test (previously in line 399), results are presented graphically in Figure 2. We have also noticed a mistake in the captions of the columns of table 2. We have amended it and we have further explained the value of Table 2. We believe that results of this part are now much easier to follow. Thanks. 

Table 1: Not properly explained; It was not mentioned in the text; What does first column represent? What is FB years?;Where are the values for Mode Column?; What is N column?

We have tried to clarify and further explain Table 1. First, we have informed the category of first column, added the mode, and a footnote with the names of abbreviations at full. We have also made sure that Table 1 is mentioned in the Results section (line 378) and have placed after first paragraph of results so that it is fully connected with the text. 

In table 2, why value of country is High and Low? What does it mean? On what basis they are considered Low and High??What is the threshold; 

2nd column should be Gender (Mistake); 

As mentioned earlier, the captions of the first and second columns in table 2 were wrong. “Country” has been substituted by “Narcissism”, “SNS use” by “Gender”, and “NPQCS” by “SNS use”. Regarding the high vs low threshold, we have added the information of the mean average and reported is as the threshold in table 2 as a footnote, in figure 2 and within the explanations in the text. 

Line 404, All countries except South Korea reported positive correlations for H1 (all p-values <.005 except South Korea’s NPQC-R) ? is this shown in the table?

Yes, indeed; we have now included the significance of data for all tables with asterisks (*) and, int eh particular case of table 3, have amended the text to which these comments refer (changing “positive” for “significant”). Also, since South-Korea does not show positive correlations between social media use and the narcissism (NPQC-R), the ANOVA model that stems from these data does not consider data from this country (table 4). 

Line 406, “Spanish teenagers reported the highest values of SNS use (Table 3), highest narcissism levels.” It is not the highest narcissism levels because South Korea is higher. 

Yes, indeed. However, as shown in the previous comment, South Korea’s data were not statistically significant; having said this, we have rephrased the original sentence in order to make clear that we refer to statistically significant evidence. 

Table 3,

in its caption, they mentioned NPI-13 but is it not in the table

How p-values are obtained? 

NPI-13 data has been added in Table 3. And as mentioned, a footnote has been added to the table so that the type of test (correlation) and significance are now reported (*). 

Line 415.” all p-values under .05, except South Korea)” please explain.

We have further clarified that South Korean adolescents did not show a significant correlation but we also highlight the fact that the other countries did show a significant correlations with narcissism. 

Line 445, “Table 6. Social media use (SNS use) and self-appearance focused behavior use of SNS”, in other table captions SNS is mentioned without the complete phrase. Consistency is required. 

We have kept the full name of variables in the titles of figures and tables (with the abbreviation in parenthesis so that the reader can easily see the equivalence when the abbreviation is used as a caption of a column or an axis of these figures and tables). Within the text, abbreviations are only used when the word is repeated in the same paragraph making sure the reader can connect it with the full word it is referring to. 

Please explain Table 4 

We have further developed the explanations that arise from Table 4 and Figure 3 with a more explicit illustration of the numbers in the table and the graphics in Figure 3. But, as explained, these explanations are limited and only reinforce previous data: the ANOVA model of Table 4 does not include South Korean because it stems from the high versus low narcissism divide as calculated with the mean of the NPQC-R index -being the case that South Korea had not shown correlation between social media use and NPQC-R. 

Figure 1 is not mentioned in the text. 

Figure 1 is highly relevant and clarifying figure in the article, so we appreciate the reviewer noticing this. The reference has been added to follow the connection of variables in the form of hypothesis and research question. 

Fig 2, what is x-axis, what is y-axis? 

We added the concepts being measured in the figure. 

Figure 4, give full form of SCA should be added. 

We added the full range of SCA and SNS use axes in figure 4.

For Figure 2,3, and 5, the threshold should be shown in the Figures 

The thresholds for the variables in the three figures have been added to the very captions of those.

---

## [Decision Letter · Decision Letter 1]

29 Jun 2022

PONE-D-21-34574R1The use of social media as a two-way mirror for narcissistic adolescents from Austria, Belgium, South-Korea, and SpainPLOS ONE

Dear Dr. Mas Manchón,

Thank you for submitting your manuscript to PLOS ONE. After careful consideration, we feel that it has merit but does not fully meet PLOS ONE’s publication criteria as it currently stands. Therefore, we invite you to submit a revised version of the manuscript that addresses the points raised during the review process. Please address the minor comments by Reviewer 2.

We look forward to receiving your revised manuscript.

Kind regards,

Rashid Mehmood, PhD

Academic Editor

PLOS ONE

Journal Requirements:

Reviewers' comments:

Reviewer's Responses to Questions

**Comments to the Author**

1. If the authors have adequately addressed your comments raised in a previous round of review and you feel that this manuscript is now acceptable for publication, you may indicate that here to bypass the “Comments to the Author” section, enter your conflict of interest statement in the “Confidential to Editor” section, and submit your "Accept" recommendation.

Reviewer #1: All comments have been addressed

Reviewer #2: (No Response)

2. Is the manuscript technically sound, and do the data support the conclusions?

Reviewer #1: Yes

Reviewer #2: Yes

3. Has the statistical analysis been performed appropriately and rigorously? 

Reviewer #1: Yes

Reviewer #2: Yes

4. Have the authors made all data underlying the findings in their manuscript fully available?

Reviewer #1: Yes

Reviewer #2: Yes

5. Is the manuscript presented in an intelligible fashion and written in standard English?

Reviewer #1: Yes

Reviewer #2: Yes

6. Review Comments to the Author

Reviewer #1: The authors have addressed all the comments that I had made. I recommend that the paper can be accepted.

Reviewer #2: The authors made progress but some of my comments are not addressed.

Comments:

- Table 1,

o still needs more explanation

What is N column? It is still not clear

o

o Line 376, “The overall mean (M) for NPQC-R was 3.6”. According to the contents in Table 1 it is 3.05??

- In the footnote for table 2, “The threshold to divide segments between high and low narcissism is the mean average of this variable, that is higher or lower than 3.05 (as also reported in Figure 2).”. Please double check.

- Table 7,

o in its caption, gender is mentioned but is it not in the table and not explained in the text

7. PLOS authors have the option to publish the peer review history of their article (what does this mean?). If published, this will include your full peer review and any attached files.

Reviewer #1: No

Reviewer #2: No

---

## [Author Response · Author response to Decision Letter 1]

30 Jun 2022

We thank the reviewer 2 for their relevant and useful comments. Reviewer #2 is addressed individually, with the reviewer’s comments in black font and our answers in blue font.

Reviewer 2

Table 1, still needs more explanation. What is N column? It is still not clear.

We see that N data is confusing in this table, so we remove it. The sample size is reported in the Method. 

Line 376, “The overall mean (M) for NPQC-R was 3.6”. According to the contents in Table 1 it is 3.05??

We have double checked and the correct figure is 3.05. It has been amended. Thanks. 

In the footnote for table 2, “The threshold to divide segments between high and low narcissism is the mean average of this variable, that is higher or lower than 3.05 (as also reported in Figure 2).”. Please double check. 

We have double checked the average of NPQC-R and it is 3,05, not 3,6.

Table 7, in its caption, gender is mentioned but is it not in the table and not explained in the text.

We have double checked. This is a mistake in the caption; “Gender” has been removed from the caption. Thanks a lot.

---

## [Decision Letter · Decision Letter 2]

28 Jul 2022

The use of social media as a two-way mirror for narcissistic adolescents from Austria, Belgium, South-Korea, and Spain

PONE-D-21-34574R2

Dear Dr. Mas Manchón,

We’re pleased to inform you that your manuscript has been judged scientifically suitable for publication and will be formally accepted for publication once it meets all outstanding technical requirements.

Kind regards,

Rashid Mehmood, PhD

Academic Editor

PLOS ONE

Additional Editor Comments (optional):

Reviewers' comments:

Reviewer's Responses to Questions

**Comments to the Author**

1. If the authors have adequately addressed your comments raised in a previous round of review and you feel that this manuscript is now acceptable for publication, you may indicate that here to bypass the “Comments to the Author” section, enter your conflict of interest statement in the “Confidential to Editor” section, and submit your "Accept" recommendation.

Reviewer #2: All comments have been addressed

2. Is the manuscript technically sound, and do the data support the conclusions?

Reviewer #2: Yes

3. Has the statistical analysis been performed appropriately and rigorously? 

Reviewer #2: Yes

4. Have the authors made all data underlying the findings in their manuscript fully available?

Reviewer #2: Yes

5. Is the manuscript presented in an intelligible fashion and written in standard English?

Reviewer #2: Yes

6. Review Comments to the Author

Reviewer #2: (No Response)

7. PLOS authors have the option to publish the peer review history of their article (what does this mean?). If published, this will include your full peer review and any attached files.

Reviewer #2: No

---

## [Editor Report · Acceptance letter]

4 Aug 2022

PONE-D-21-34574R2 

The use of social media as a two-way mirror for narcissistic adolescents from Austria, Belgium, South-Korea, and Spain 

Dear Dr. Mas Manchón:

I'm pleased to inform you that your manuscript has been deemed suitable for publication in PLOS ONE. Congratulations! Your manuscript is now with our production department. 

Kind regards, 

on behalf of

Dr. Rashid Mehmood 

Academic Editor

PLOS ONE